# A Greek Parliament Proceedings Dataset for Computational Linguistics and Political Analysis

Konstantina Dritsa[1], Kaiti Thoma[2], John Pavlopoulos[3], and Panos Louridas[4]

[1,2,3,4]Athens University of Economics & Business, Greece, [3] Stockholm University, Sweden,
[1]dritsakon@aueb.gr, [2]aikelthoma@gmail.com, [3]annis@aueb.gr, [4]louridas@aueb.gr

## Abstract

Large, diachronic datasets of political discourse are hard to come across, especially for resource-lean languages such as Greek. In this paper, we introduce a curated dataset of the Greek Parliament Proceedings that extends chronologically from 1989 up to 2020. It consists of more than 1 million speeches with extensive metadata, extracted from 5,355 parliamentary record files. We explain how it was constructed and the challenges that we had to overcome. The dataset can be used for both computational linguistics and political analysis—ideally, combining the two. We present such an application, showing (i) how the dataset can be used to study the change of word usage through time, (ii) between significant historical events and political parties, (iii) by evaluating and employing algorithms for detecting semantic shifts.

## 1 Introduction

The meanings of words change continuously through time, reflecting the evolution of societies and ideas. For example the word "gay" originally meant "joyful" but gradually changed its usage to refer to sexual orientation [22]. Perhaps less well-known, but probably more relevant to our subject, in 1850 rubbish-tip pickers were using the term "soft-ware" for material that will decompose and "hard-ware" for the rest [8, p. 380]. The availability of large corpora and advances in computational semantics have formed fertile ground for the study of semantic shifts. When these advances are applied to parliamentary corpora, they can provide useful insights into language change[1], different political views, and the effect of historical events to the use of a language.

In this paper, we present a curated diachronic Greek language dataset, extracted from the proceedings of the Greek Parliament and spanning 31 years. It consists of more than 1 million speeches in chronological debate order, with extensive metadata about the speakers, such as gender, political affiliation, and political role. To our knowledge, it is the only freely available dataset covering a comparable length of time in the Greek language. Moreover, by its nature as a record of the country's parliament, it is again to our knowledge the only dataset that captures more than a quarter century of the recent Greek political history. We demonstrate the value of the dataset by using it to evaluate four state of the art word usage change detection approaches and select the most appropriate among them to compute word usage changes across time and among political parties.

The paper is organized as follows: Section 2 summarizes the approaches for word usage change detection. Section 3 presents our dataset and its construction process. In Section 4 we evaluate four state of the art word usage change detection algorithms. In Section 5 we examine how word usage changes reflect political events in Greece. Section 6 presents our conclusions and further discussion.

---

[1]"Language evolution", "lexical semantic change", "terminology evolution", "semantic change", "semantic shift", and "semantic drift" are also all terms used for the same concept.

36th Conference on Neural Information Processing Systems (NeurIPS 2022) Track on Datasets and Benchmarks.

## 2   Related Work, Challenges, and the Parliamentary Dataset

Researchers have attempted to capture diachronic semantic shifts of words with the use of distributional semantics, based on the distributional hypothesis [13]. According to this hypothesis, each word is characterized by the company it keeps. It follows that the change in the usage of word, that is, its semantic shift, is defined by the change in the words co-occurring in its context. Computationally, words are embedded in short dense vectors according to their co-occurrence relationships and word usage change can be measured by the distance between vectors that are calculated on data of different time periods [3]. Approaches of capturing diachronic semantic shifts can be divided into projection-based and neighbor-based [16, 26]. The former have shown to be mostly suitable for detecting changes of linguistic drift, more prominent in verbs, while the latter for capturing cultural semantic shifts, encountered more frequently in the nominal domain [26, 17].

According to the projection-based approach, word vectors calculated on different corpora are projected in a shared vector space and usage change is computed with the cosine distance. However, vectors trained on different corpora are not comparable by default, as word embedding algorithms are inherently stochastic. Thus, many transformation methods have emerged, with the most prominent being vector space alignment [18, 34, 22, 21, 38]. Hamilton et al. [18] (hereafter **"Orthogonal Procrustes"**) use orthogonal Procrustes transformations to align diachronic models. Recent studies are still building on this work: such is the case of incremental update methods, where one trains a model on one corpus and then updates it with data from the other corpus, while saving its state every time [21, 38].. Carlo et al. [7] (hereafter **"Compass"**) build on the assumption that it is the context of the word that changes over time, but the meanings of the individual words in each different context remains relatively stable. From that assumption they work with the context embeddings learned by word2vec models[33], trained on atemporal target embeddings that function as an alignment compass.

The neighbor-based approach uses directly the different neighboring words that reflect change. Gonen et al. [16] (hereafter **"NN"**) introduce intersection@$k$, i.e., the intersection of the word's top-$k$ nearest neighbors from each corpus, to measure the difference of neighboring words. In their work, they propose that projection-based methods are more sensitive to proper nouns.

Hybrid approaches have emerged, combining properties of the aforementioned categories. Hamilton et al. [17] (hereafter **"Second-Order Similarity"**) collect the word union of the top-$k$ neighbors of a word $w$ from two different corpora. Then, they create a second-order embedding for each corpus with the similarity between $w$ and each neighbor. Intuitively, usage change is estimated by the angle the word's neighborhood has to cover when moving from one corpus to the other. In their work they propose that cultural changes should be studied with neighbor-based approaches.

Furthermore, the rise of contextual embeddings such as BERT [6] and ELMo [39] has enabled important developments in the study of word usage change as they are capable of generating a different vector representation for each specific word usage. Contextual embeddings can be used in the context of usage change detection by aggregating the information from the set of token embeddings [35, 29, 30, 24, 15]. However, related work shows that, for the time being, it is complex to disambiguate between word senses, and there is a large disparity between results on different corpora [30, 29, 35, 27]. Finally, recent studies have emerged that ensemble multiple types of word embeddings and distance metrics to experiment on improving overall performance [24, 31].

Different approaches can give different results, thus comparing them is a challenge [43]. An additional challenge is the stability of the approach used. An approach demonstrates stability when slightly different runs on a dataset do not significantly affect the results [16]. Recent studies highlight the importance of stability, as a high variation can be a sufficient reason to call the whole method into question [2, 4]. Researchers have identified a number of factors that affect stability, including properties of the underlying algorithms used to construct the embeddings [16, 46, 2, 28, 19]. Subsequent runs of word embedding algorithms on the same data will not necessarily produce the same results, due to the stochastic nature of the approaches. Gonen et al. [16] use intersection@$k$, mentioned above, to gauge the stability between the predictions of two different runs of the same algorithm. We adopted this metric, in order to select a stable usage change algorithm for our study.

Existing studies on language change use corpora of high resource languages such as English, German, French, Spanish, and Chinese, spanning centuries [41, 1, 5] or decades, comprising tweets and product reviews [22], digital books [32] and news articles [42, 37]. In English, a work similar to ours is that of Azarbonyad et al. [3], in which they study the semantic shift of words in the British House

of Commons. Also in English, Gentzkow et al. [14] curated a dataset of the US Congress speeches from 1873–2017, with extensive metadata on speeches and speakers.

In this work, we present an extensive dataset that can be used for the study of language change in the context of the Greek Parliament. To the best of our knowledge, there are no existing computational studies on language change in modern Greek. We show the value of the dataset by utilizing it to comparing four state-of-the-art approaches of language change detection, namely Orthogonal Procrustes [18], Compass [7], NN [16] and Second-Order Similarity [17]. The selection of the approaches for language change detection aims to be representative of different established methodologies proposed in the related work and does not constitute a complete benchmark evaluation on language change detection methods. Furthermore, following the challenges identified above, we evaluate the stability of the approaches using the intersection@$k$ measure. We also qualitatively evaluate their results on the change of word usage between the decades 1990–1999 and 2010–2019. Finally, as the dataset is a mirror of political history, we use it to detect word usage changes between different time periods, before and after important historical events, as well as among different political entities.

## 3 Dataset Description and Construction

### 3.1 Contents

The dataset[2] includes 1,280,918 speeches of parliament members in chronological debate order, exported from 5,355 parliamentary sitting record files, with a total volume of 2.12 GB. The speeches extend chronologically from July 1989 up to July 2020. Table 1 shows the contents of the dataset.

| | |
|---|---|
| **member_name** | the name of the person speaking |
| **sitting_date** | the date the sitting took place |
| **parliamentary_period** | the name and/or number of the parliamentary period that the speech took place in. A parliamentary period is defined as the time span between one general election and the next. A parliamentary period includes multiple parliamentary sessions. |
| **parliamentary_session** | the name and/or number of the parliamentary session when the speech took place. A session is a time span of usually 10 months within a parliamentary period during which the parliament can convene and function as stipulated by the constitution. A parliamentary session includes multiple parliamentary sittings. |
| **parliamentary_sitting** | the name and/or number of the parliamentary sitting that the speech took place in. A sitting is a meeting of parliament members. |
| **political_party** | the political party of the speaker |
| **government** | the government in power when the speech took place |
| **member_region** | the electoral district the speaker belonged to |
| **roles** | information about the speaker's parliamentary and/or government roles |
| **member_gender** | the gender of the speaker |
| **speech** | the speech delivered during the parliamentary sitting |

Table 1: Contents of the Parliament Proceedings Dataset

Delving deeper into our dataset, Fig. 1 depicts the percentage of female members in the Greek Parliament and the percentage of characters of speech delivered by female individuals, per political party and per parliamentary period. The difference between the membership percentage and the speech percentage is highlighted with dotted vertical lines. For reasons of readability, we depict political parties that have played an important role in recent political history. These are New Democracy (center-right, hereafter ND), the Panhellenic Socialist Movement (center-left, hereafter PASOK), the Coalition of the Radical Left—Progressive Alliance (left, hereafter SYRIZA), the Communist Party of Greece (communist, hereafter KKE), the Coalition of the Left, of Movements and Ecology (left, hereafter SYN) and Golden Dawn (extreme right, nationalist, nazi-fascist, hereafter GD). We exclude period 14, which lasted two days and was a transitional government.

Concerning gender representation in the parliament, the total percentage of female members (dashed line) increases over time. Left-wing political parties like SYN, KKE and SYRIZA achieve higher

---

[2]https://zenodo.org/record/7005201

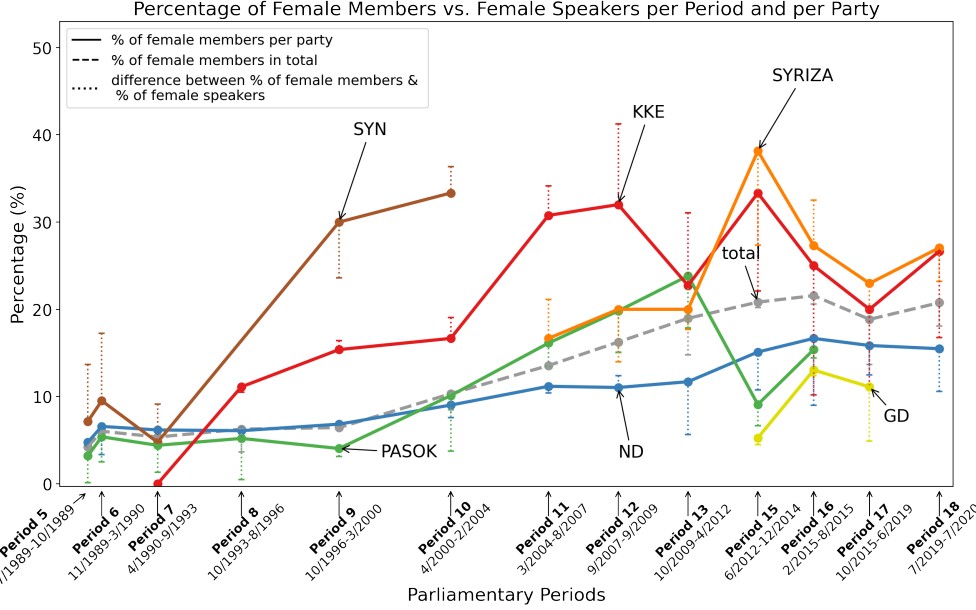

Figure 1: Percentage of female members and corresponding speech activity per period.

percentages of female members and remain above the total percentage of female members for almost all periods. The percentage of center-left PASOK presents great fluctuation over the years. On the other hand, the percentage of the center-right ND remains below the total average percentage. Lastly, the far-right GD has the lowest percentage of female members of parliament, compared to the selected political parties. Regarding the participation of females in parliamentary debates, only the left-wing SYN and KKE achieve percentages higher than that of female membership for most periods. Overall, none of the examined parties has a percentage of female members equal to or greater than 50% at any point in time. After investigating the rest of the parties, we found that only two left-wing parties have achieved percentages greater than 50% for female members, namely Alternative Ecologists (greens, left, 100% for periods 6 & 7) and MeRA25 (left, 55% for periods 16 & 18).

## 3.2 Record Collection

Due to the absence of an API, we crawled the catalogue of parliament records from 1989 up to 2020, available from the official Greek Parliament website[3]. Files were in doc, docx, text, PDF or HTML format. We converted all to text using Apache Tika[4].

Each record of a parliamentary sitting begins with some introductory information, followed by the debate that took place. Typically, each speaker's full name is written in capital letters at the beginning of a new line, followed by a colon and the corresponding speech. The name is occasionally accompanied by a parenthesis with information about the person, such as their political party or governmental role. Unfortunately, the records contain material that fails to follow this format. So, to extract speeches from the parliamentary records it was necessary to create, in a preliminary step, a number of auxiliary datasets as described below.

## 3.3 Support Datasets

**Female & Male Names** We crawled the entries of the Wiktionary Greek names category[5] and created a support dataset of modern Greek female and male names and surnames and their grammatical cases, filling missing entries using the rules of grammar.

---

[3] `https://www.hellenicparliament.gr/Praktika/Synedriaseis-Olomeleias`. The proceedings for 1995 are not publicly available.

[4] `https://tika.apache.org/download.html`, tika-app-1.20.jar

[5] `https://en.wiktionary.org/wiki/Category:Greek_names`

**Elected Members of Parliament**    The Greek Parliament website provides a list[6] of all the elected members of parliament since the fall of the military junta in Greece, in 1974. For each member, we extracted the exact date range of their activity in each political party during each parliamentary period. We added the gender of each member, based on the gender of their name from the "Female & Male Names" dataset.

**Government Members**    As government members we refer to individuals in ministerial or other government posts, regardless of whether they were elected in the parliament. This information is available in the website of the Secretariat General for Legal and Parliamentary Affairs[7]. Names and surnames are given in the genitive case and cannot be matched directly to parliamentary records, where names are given in the nominative case. To resolve this, we used the "Female & Male Names" dataset to convert the collected genitive cases to nominative and deduce the gender.

**Governments**    We automatically collected from the website of the Secretariat General for Legal and Parliamentary Affairs[7] a support dataset with the names of governments since 1989, their start and end dates, and a URL that points to the respective official government web page of each government.

**Additional Political Posts**    We manually collected from Wikipedia additional government and political posts that were not included in the previous resources: service information of the Chairmen, Speakers and Deputy Speakers of the Parliament, party leaders, and opposition leaders.

**Merged Support Dataset**    We merged the above datasets producing an integrated file. Each row of the final file includes the full name of the individual, the start and end date of their term of office, the political party and electoral district they belonged to, their gender, the parliamentary and/or government positions that they held along with start and end dates, and the name of the government that was in power during their term of office.

### 3.4    Speech Extraction

**Speaker Detection**    To identify each new speech, we had to identify a valid candidate speaker. As mentioned, in many cases the text did not follow the expected format. For example, some new speeches would not start at the beginning of a new line or there would be missing closing brackets in the speaker's reference. We created a comprehensive list of regular expressions in order to capture possible debate formats.

**Entity Resolution**    After the detection of a candidate speaker, we matched the extracted speaker to our list of individuals with the use of the Jaro-Winkler [47] string similarity metric. However, although not as problematic as characters in a Russian novel, there exist many different name variants in the records. Some speakers were referenced with their nicknames. For people with more than one names/surnames, some of them where missing and the order of the first/last names was not always the same. To resolve this string comparison task, we created all possible variants of an official name, alternating the order of the words that make up that name and replacing or combining the name with its variants. Due to misspellings, we accepted matches with similarity $\geq 0.95$. For matching we used yet another dataset of 475 names and nicknames, which cannot be shared due to licensing reasons.

### 3.5    Preprocessing

We replaced all references to political parties with the symbol "@" followed by an abbreviation of the party name. We removed accents, strings with length less than 2 characters, all punctuation except full stops, and replaced stopwords with "@sw".

The volume of data per parliamentary period varies, as does the shared vocabulary between consecutive periods. This is key to our investigation, as commonalities between the vocabularies across time are necessary to detect usage change. Fig. 2 shows the common vocabulary in terms of tokens between pairs of consecutive periods. Periods 5, 6, 14 and 16 are transitional and span between 1 to 7 months, resulting in low vocabulary overlap with other periods. In these cases of small shared vocabulary, important semantic shifts are usually artifacts of the lack of data. To avoid biased conclusions, we merged the smaller periods with their following large period, these being period 5 and 6 with period 7, period 14 with 15 and period 16 with 17. Table 2 shows descriptive statistics of the dataset in three different steps, before and after preprocessing, and upon preprocessing and merging the smaller

---

[6] https://www.hellenicparliament.gr/Vouleftes/Diatelesantes-Vouleftes-Apo-Ti-Metapolitefsi-Os-Simera/

[7] https://gslegal.gov.gr

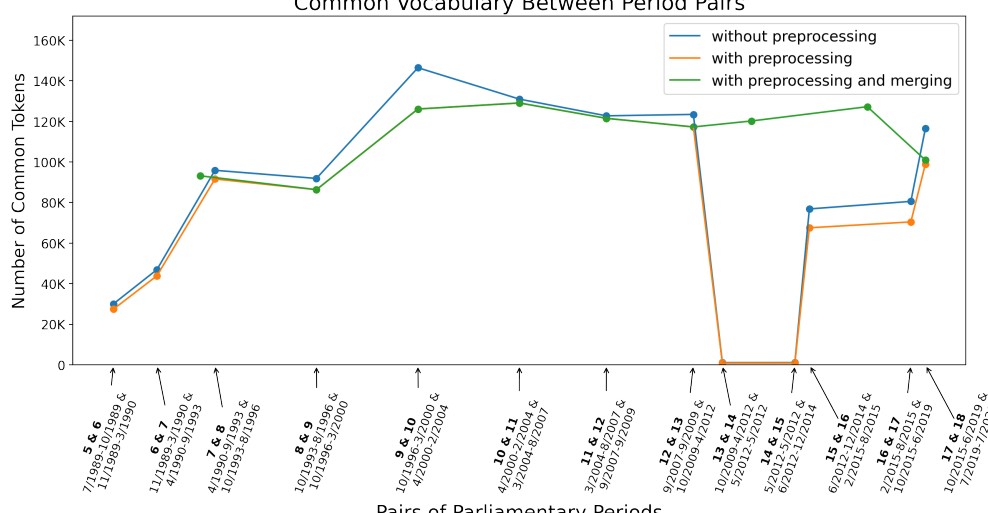

Figure 2: Common vocabulary between consecutive pairs of periods before preprocessing, after preprocessing and upon preprocessing and merging small periods with the next one.

Table 2: Average metrics for each parliamentary period before preprocessing, after preprocessing, and after merging the small periods with their consecutive large periods.

|  | Avg. characters | Avg. tokens | Avg. unique tokens | Avg. sentences | Avg. unique sentences |
|---|---|---|---|---|---|
| **No preprocessing** | 82.4M | 14.18M | 208.12K | 570.11K | 516.99K |
| **With preprocessing** | 78.22M | 19.57M | 216.79K | 416.46K | 386.83K |
| **With preprocessing & merged periods** | 109.51M | 27.4M | 287.19K | 583.04K | 541.04K |

periods with their following large periods. Preprocessing leads to a decrease of the average number of characters and sentences but increases the average tokens and unique tokens. Merging the periods increases all numbers.

## 4 Quantitative and Qualitative Evaluation

### 4.1 Quantitative Evaluation: Stability

We compared stability between Orthogonal Proctustes [18], Compass [7], NN [16] and Second-Order Similarity [17], as well as a variation of the Compass method in which we introduced the frequency cut-offs of the NN approach [16]. Specifically, we removed from the vocabulary of each model the 200 most frequent words and words that appear less than 200 times. In our case, the frequency distribution for each corpus is long-tailed, with only ∼5% of the vocabulary of each decade having 200 or more occurrences. Our aim is to investigate whether the removal of these words might increase the stability of the results.

We applied the comparison on the task of word usage change detection between 1990–1999 and 2010–2019. We used intersection@$k$, proposed by Gonen et al. [16], which measures the percentage of shared words in the $k$ most changed words for a number of restarts, each time changing the random seed. For each approach (e.g., Compass) and between the two time periods, we measured word usage change and detected the most changed words. By repeating the measurement with different random seeds, then, we computed the common changed words across repetitions. Specifically, we ran each usage change approach 10 times and collected the top-$k$ most changed words, where $k \in [10, 20, 50, 100, 200, 500, 1000]$. Then, for each of the $\binom{10}{2} = 45$ pairs of different runs and for each of the values of $k$, we measured the percentage of shared words in the most-changed-words predictions of each approach. A value of zero between a pair of runs means there are no shared words in their predictions, indicating high variability, while a value of one indicates high stability.

Fig. 3 shows the average intersection@$k$ between the 45 pairs of different runs for each approach, with 95% confidence intervals calculated by the bootstrap method. The NN method exhibits the greatest stability even for very small values of $k$. This means that for all values of $k$ and regardless of

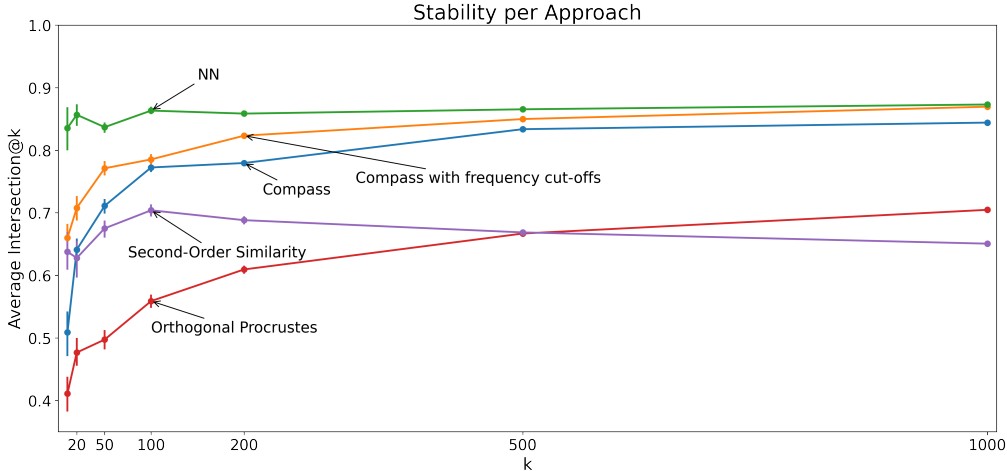

Figure 3: The average intersection@$k$ for 45 pairs of different runs and for different values of $k$.

Table 3: A representative selection of words from the top 100 most changed words per approach between the 1990s and 2010s.

|  | Selection of top changed words |
|---|---|
| **Compass** | psi, haircut, normality, vatopedi, cook |
| **Compass variation** | agenda, inputs, european economic community, drachma, green |
| **NN** | simple, deny, called, people, interested |
| **Orthogonal Procrustes** | red, clarity, capital, Prespa, migratory |
| **Second-Order Similarity** | give, phthiotis, laconia, arcadia, critical |

the random seed used, the changed words this approach detected were mostly the same. Compass follows closely, while the Compass variation yields better stability results, similar to that of NN. The Orthogonal Procrustes and the Second-Order Similarity approach gave worse stability results, with the latter even decreasing for $k > 100$.

## 4.2 Qualitative Evaluation: Top Changed Words between 1990-1999 & 2010-2019

We qualitatively evaluated the top 100 most changed words between the decades 1990–1999 and 2010–2019, as detected by each approach. We introduce a frequency threshold of 50 occurrences in at least one of the two decades, for the approaches that do not already have frequency thresholds. Table 3, shows a representative selection of results for each approach.

Compass detected words that have meaningful change connected with Greek historical events. The words "psi", "haircut", "normality", "agenda", "drachma" are largely related to the Greek financial crisis of the 2010s. "PSI" stands for "Private-Sector Involvement", meaning that private investors had to accept a write off on the face value of Greek government bonds they were holding. A "haircut" is a cut to existing debt. "Vatopedi" referred to an economic scandal involving an homonymous monastery. "Cook" was connected to the bankruptcy of the British travel firm "Thomas Cook", possibly affecting the Greek tourist industry.

Orthogonal Procrustes also detected word usage changes that make sense in the historical context. "Red" was used for the so-called "red loans", non-performing loans that emerged during the crisis. "Clarity" became connected to the online platform `diavgeia.gov.gr`, where government spending is publicly published to improve transparency. "Capital" in the 2010s referred to the capital controls applied in the Greek banks. "Prespa", name of a lake, was used in the 2010s to describe the Prespa agreement between Greece and the Republic of North Macedonia. Change in the word "migratory" reflects the increased arrivals of refugees by sea in the 2010s, mainly due to the Syrian civil war.

NN and Second-Order Similarity approaches provided less explainable results. The top-100 list of NN included mainly verbs and no proper nouns. The closest neighbors of the verbs consisted mainly of grammatical persons, tenses and synonyms. The Second-Order Similarity results included almost only geographical regions, numbers and names of months.

# 5 Changes in Word Usage and Political History

As Compass presented the best performance combination in both stability and detection of meaningful change, we used it to investigate changes in word usage and events in Greece's recent political history.

## 5.1 Top Changed Words before and after the Greek Economic Crisis

Greece faced a threat of sovereign default in 2007–2008, leading to a massive recession. In the following years, Greek governments adopted austerity measures in a series of adjustment programs agreed with Eurozone countries and the International Monetary Fund (IMF). We detect word usage changes between the decades before ($t1$: 1997–2007) and during ($t2$: 2008–2018) the crisis.

Table 4: Words with notable usage change before ($t1$) and during ($t2$) the Greek economic crisis.

| Word | Similarity | Neighbors @ $t1$ | Neighbors @ $t2$ |
|---|---|---|---|
| haircut | -0.06 | gypsy, sixteen-year-old, excellent, empirical | psi, repurchase, haircut, reduction |
| psi | 0.01 | boilers, rented, fainted, humanization | haircut, repurchase, bonds, sector |
| golden | 0.01 | feed, renegotiate, people, pretend | boys, platinum, chicago, hall |
| story | 0.01 | tortures, old-fashioned, nail, tobaccoworker | success, true, fairy tale, myth |
| success | 0.04 | dried, liberated, interbank, emerging | story, myth, make up, fairy tale |
| brain | 0.05 | distinguished, overpay, collected, dermatological | drain, gain, circulation, migration |
| cutter | 0.06 | tweaked, fighting, salvage, rescuing | automatic, mechanism, account, infamous |
| systemic | 0.06 | autumn, short-term therapy, shape, segmented | corrupted, media, unchecked, regime |
| imf | 0.06 | hall, bleed, multivariate, superset | schäuble, troika, monetary, european commission |
| counter-measures | 0.10 | resin, legal policies, fainted, peaks | burdensome, painful, anti-popular, recessionary |

Table 4 presents a selection of words with notable word usage change, along with their closest neighbors, at each of the two decades; as usage change is measured with the cosine similarity, low values represent significant change. "Haircut" did not initially refer to "a debt haircut"; PSI was a unit of pressure in $t1$, before referring to private sector involvement in Greek bonds write off. "Cutter" was used to describe measures for economic stability, such as unemployment allowances. "Golden" in $t2$ became part of the phrase "golden boys", referring to people working in senior management positions with high incomes and provocative lifestyles, whose administrative decisions usually burdened their companies. It was also used in references that liken the crisis policies with those of the Chicago Boys, the Chilean economists of the Pinochet rule educated at the University of Chicago. "Success" and "story" were regularly employed together to ironically describe government promises of economic prosperity. "Brain", appearing in "brain drain", referred to the migration of highly skilled people to other countries in search for better living conditions. The word "systemic" was commonly used to negatively characterize mainstream media that, while heavily indebted themselves, were supporting government policies. "IMF" was used during the crisis in the context of the strict financial reforms it required from the Greek government. "Countermeasures" referred to government's compensatory measures against the economic austerity.

## 5.2 Usage Change of Popular Topics

We estimate the usage change of selected topics that were debated across periods. For the selection of topics, we consulted the website of Vouliwatch[8], a non-partisan parliamentary monitoring organization that provides an extensive comparison of party positions on 69 topics of significant political interest. We extended this list with 22 additional topics, selected for their popularity[9]. We repeated the usage change computations with 50 different random seeds and calculated 95% confidence intervals with the bootstrap method.

Fig. 4 shows a subset of topic embeddings that exhibit notable decrease in semantic similarity ($\leq 0.65$) over at least one pair of consecutive periods. The word "macedonian" changes around 2000 from referring to a business consortium named "Macedonian metro", to portraying the turbulence around the naming dispute between Greece and the Republic of North Macedonia. The similarity drop between periods 14–15 and 16–17 corresponds to extensive debates that took place in the parliament nearing the dispute resolution in 2019. The word "refugee" changed from referring to cheap labor to reflecting the increased number of persons crossing the borders to seek asylum in the EU. The notable drop around 2015 is in agreement with external data recording an increased

---

[8] https://vouliwatch.gr
[9] The initial and extended lists of topics are available in the supplementary material of the paper.

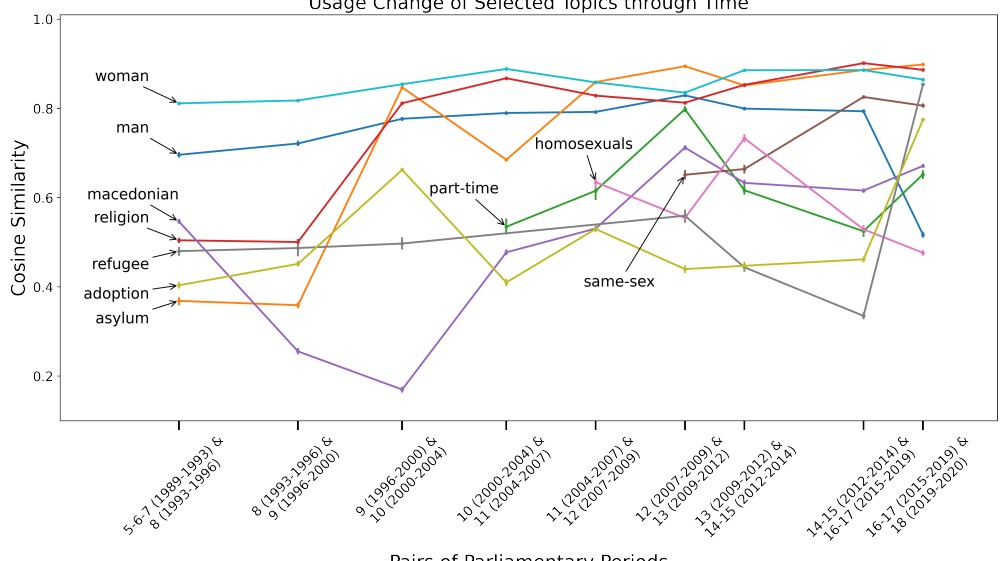

Figure 4: Usage change of 10 political topics between pairs of parliamentary periods. Low cosine similarity denotes high usage change.

volume of refugee arrivals at the time [10]. During the economic crisis, from 2008 onwards, "part-time" changed usage as it became a common employment practice to reduce salary expenses. The terms "homosexuals" and "same-sex" emerge around 2007, reflecting an increasing social awareness. In the following years, the terms undergo important usage change as they approach the words "marriage", "cohabitation agreement" and "adoption". The word "man" changes context in 2019 from describing a male of the typical Greek family model or a criminal to referring to a policeman, associated with arbitrary police behavior and brutality. The word "woman" does not display notable usage change but we include it for comparison with the word "man". "Woman" is constantly correlated with the words "mother", "child", "spouse", "family", exhibiting a context limited to traditional family relations.

### 5.3 Usage Change of Political Party Name Embeddings

We gauge the usage change between parliamentary periods of the names of political parties that have played an important role in recent political history, introduced in Section 3.1.

As mentioned in Section 3.4, we replaced all political party references with the symbol "@" followed by an abbreviation of the party name, using regular expressions that capture grammatical cases and variations. We trained Compass between consecutive pairs of parliamentary periods and computed the cosine similarity between the vectors of political party names. We repeated the computations with 50 different random seeds and calculated 95% confidence intervals with the bootstrap method. Fig. 5 presents the results. References to political parties in the records through time do not reflect their actual life-cycle. For example, although SYN was dissolved in 2013, references to it persist in the following years. ND, PASOK, and KKE show high similarity scores between all pairs of consecutive parliamentary periods, reflecting a stable political position. We locate the period pairs for which each party embedding shows the lowest semantic similarity and study its neighbors for each period to shed more detail to their usage change. During 2012–2014, ND appears closer to the words "opposition" and PASOK, as it was the official opposition party of the government of PASOK. It is also close to the word "Karamanlis", the name of the party leader at the time. During 2015–2019, ND comes closer to the words "coalition government", PASOK and "DIMAR", an abbreviation of the Democratic Left party, a minor left-wing political party not shown in Fig. 5. This change in usage is consistent with political events of the period, when ND formed a coalition government with PASOK and DIMAR. The usage change of PASOK between the periods 2015–2019 and 2019–2020 coincides with the incorporation of PASOK as the basic component of a new political party, KINAL. The lower cosine similarity for KKE between 1989–1993 and 1993–1996 reflects the multiple coalitions and divisions it went through at these periods (also reflecting effects in global history), after which it

---

[10]https://data.unhcr.org/en/situations/mediterranean/location/5179

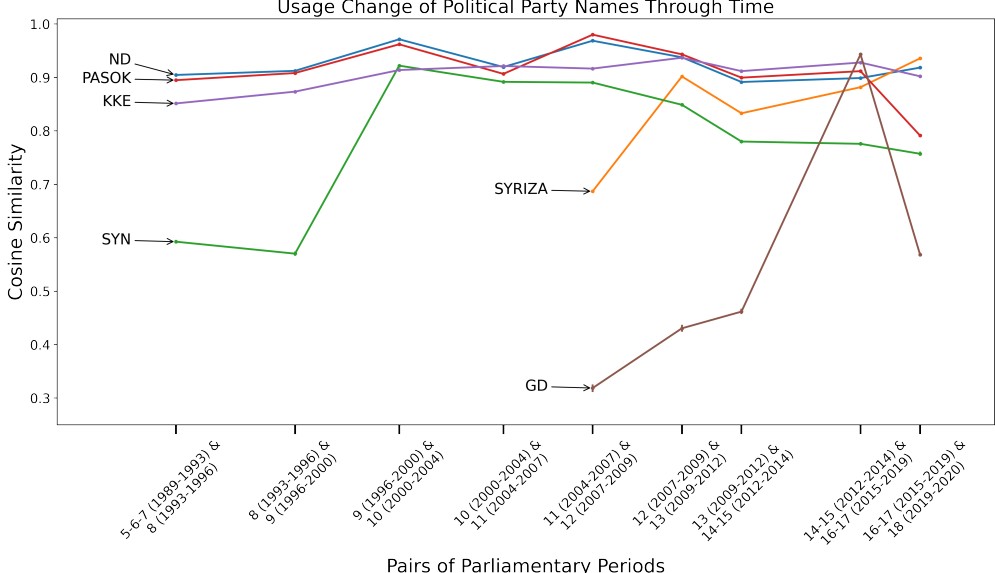

Figure 5: Usage change of political party embeddings between pairs of consecutive periods.

remained in a stable state. For SYN, the usage change between the periods 1993–1996 and 1996–2000 coincides with a crisis brought by a failure to enter parliament, the resignation of the party leader, and the election of new leadership. During 2004–2007, SYRIZA is highly correlated with the name "Alavanos", the then party leader, as well as SYN, the largest component of the alliance that constituted SYRIZA. In 2007, Alavanos was succeeded by Tsipras. In the following years, SYRIZA evolved from a loosely-knit coalition to the largest party in parliament in 2015, leading a government coalition with a minor partner (Independent Greeks, ANEL) in 2015–2019. That is mirrored by highest cosine similarity, perhaps echoing a consistent anti-austerity and anti-neoliberal message. GD rose to prominence during the financial crisis. During its period in the sun, GD was close to words like "brutal", "beatings", "anarchist", "marches", "episodes" and "abusive", reflecting the criminal acts and attacks that perpetuated by supporters, members, and high-ranking cadres of GD. Support for GD nosedived and did not reach the 3% threshold required to enter parliament in 2019.

## 6 Conclusions and Future Work

Large datasets of resource-lean languages on specific domains are hard to find. In this work, we present a dataset of the Greek Parliament proceedings spanning 31 years and consisting of more than 1 million speeches, tagged with extensive metadata, such as speaker name, gender and political role. We apply stable semantic shift detection algorithms and detect notable word usage changes connected with historical events such as the Greek economic crisis as well as changes in the usage of political party names, connected with internal organizational changes or election periods.

Our dataset has a specific provenance, parliamentary recordings, and is not necessarily representative of language use and evolution in general. Yet, it can be useful in various applications of computational linguistics and political science, e.g., studies that examine whether word usage change behaves differently in different languages or contexts. Its extensive metadata can facilitate fine-grained semantic change studies, such as to evaluate whether a new parliament member gradually adjusts their speech to the style of the majority of speakers. It can be used for monitoring and tracking events and controversial topics over time [25, 20, 45, 10, 9] as well as rapid discourse changes during crisis events [44], or for classification of political texts [23]. Other applications can include political perspective detection [49] and viewpoint analysis [3] between parties, roles or genders and cross-perspective opinion mining [11, 40]. The dataset can be combined with datasets of tweets and public statements of parliament members, for modeling voting behavior and improving the tasks of roll call vote and entity stance prediction [36, 12, 48].

**Supplementary Material** Download links to the original proceeding records, source code files and implementation details are included in the supplementary material that accompanies this paper. The repository for this work is `https://github.com/Dritsa-Konstantina/greparl`.

**Acknowledgments** This work was supported by the European Union's Horizon 2020 research and innovation program "FASTEN" under grant agreement No 825328 and the non-profit data journalism organization iMEdD.org.

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
