# OpenReview forum: "A Greek Parliament Proceedings Dataset for Computational Linguistics and Political Analysis"
_NeurIPS.cc/2022/Track/Datasets_and_Benchmarks — NeurIPS 2022 Datasets and Benchmarks _

### Official Review · Reviewer_pana · 2022-07-06
**Interesting data but limited theoretical insights**

**Rating:** 4
**Confidence:** 3

**Strengths:**

The dataset is new, in a relatively under-covered language, and with some useful particular properties. The collection of metadata is very rich and would allow substantive follow-up studies.

**Weaknesses:**

The applicability of the dataset beyond this task, or perhaps other model eval tasks for the same language, was unclear. The authors provided little explanation for why we should expect some models to perform better or worse in this setting. The authors picked some performance metrics without adequately explaining why we should care about them. The substantive conclusions seemed like post-hoc interpretations with no theoretical grounding.

**Additional Feedback:**

N/A

**Clarity:**

The paper was relatively clear. The exposition could be improved by stating assumptions more clearly, making more effort to provide theoretical grounding for the choice of benchmarks and performance metrics. Uncertainty in the estimates should be more clearly documented.

**Correctness:**

The dataset appears to be constructed soundly, though as noted above the removal of stopwords seems like a somewhat strange choice. I would recommend that the authors make available an unfiltered version of the data so that it would be usable in settings were stopwords should not be eliminated.

**Documentation:**

This is public data about public figures so it seemed OK.

**Ethics:**

I saw none.

**Relation To Prior Work:**

Yes.

**Summary And Contributions:**

Congratulations to the authors for assembling an interesting dataset that will be useful for benchmarking models in Greek. It clearly took a lot of effort to clean and assemble the data. And I learned many interesting things in the paper about Greek political history. Thank you for the opportunity to read it. The supplementary materials are clear and the data appears to be in good shape.

Unfortunately, I do not think this paper meets the standards for acceptance at NeurIPS. Because there is little theoretical discussion of how Greek or the particular setting of parliamentary speech is different from other settings where semantic change over time has been studied, it might be harder for others to incorporate the data into their work. Providing more intuition on why we should expect this data to reveal would be useful. A clearer theoretical grounding for the differences between methods would help motivate the results. And the substantive discussion in s5 felt very much like post-hoc explanation rather than testing hypotheses generated ex ante.

Further, there were some confusing assumptions made that were not adequately explained. For example, significant space is devoted to testing the stability of various approaches, but it is not obvious why we should prize variance over bias, especially given that the most stable method appears to yield the worst substantive results. The authors do not adequately explain the different performance measures the review and experiments on semantic change were less illuminating. The decision to remove stopwords was somewhat surprising since my understanding is that at least some approaches depend on stopwords to pin down projections. And the substantive discussion in figs 3 and 4 was strange, as some of the words seemed to be constantly changing even though the authors only pointed to some discrete changes (e.g., refugee appears to be constantly changing with a discontinuous jump in stability in the final period but the authors don't really have a strong explanation beyond gradual changes through the 2010s; it would be nice to have some real data on the volume of arrivals to explain the shape of the curve). Generally it would be more helpful to have some substantive measure ('similarity to X') rather than just plotting change so that the intuition would be clearer. A final tiny point is to please include confidence intervals especially where you are already sampling seeds.

Overall, I thought the data seems like a nice contribution but the paper did not do enough to really demonstrate the more general value of the data to a broader audience.

---

> ### Author Response · Authors · 2022-08-29
> **Response - Part 1**
>
> Thank you for your kind comments.
>
> Regarding your valid concern about stopword removal, we followed the steps proposed by related work. Stopword removal is implemented in both the neighbor-based approach of Gonen et. al. [16] as well as the projection-based approach of Hamilton et. al. [18]. Intuitively, we understand that there are many words apart from the most common stopwords that can act as anchors to align vector spaces. However, it would be interesting to study in the future if and how the removal of stopwords might affect the performance of projection-based methodologies.
>
> Furthermore, after your proposal to make available an unfiltered version of the data, usable in settings were stopwords should not be eliminated, we have uploaded a new version for our Zenodo repository here https://zenodo.org/record/7005201#.Yv4mJcFBxBI. In this new version, we created a subfolder called "dataset_versions" where the following versions of the dataset are included:
>
> - "tell_all.csv": The initial file of all extracted speeches, before preprocessing and cleaning.
> - "tell_all_FILLED.csv": This file is an intermediate version that includes improvements in the consistency and completeness of the data. Specifically, this file is produced by filling the missing names of chairmen of various parliamentary sittings of the "tell_all.csv".
> - "tell_all_cleaned.csv": The final version of the dataset that has been further cleaned and preprocessed with the use of the script “corpus_preprocessing.ipynb” and is used for our word usage change study.
>
> The files "tell_all.csv" and "tell_all_FILLED.csv" were already available in a supplementary Zenodo upload (https://zenodo.org/record/6644941#.Yv1Zz8FBzZU), as pointed out in Section 2 of the "supplementary_material.pdf". In the same upload, we provide the following two subfolders, with all the downloaded record files from the Greek Parliament website:
>
> - "original_data": A folder of the original record files downloaded from the website of the Greek Parliament. The filenames are edited to follow the naming format "recordDate_id_periodNo_sessionNo_sittingNo.ext".
> - "_data": A folder of the record files converted to text format with file names translated to English.
>
> In order to make it easier for readers to incorporate our dataset into their work and better demonstrate its general value, as per your proposal, we have enriched the Section 6 “Conclusions and Future Work” with the interesting questions you raise as well as many other applications for which our dataset can prove useful. We have also included the interesting question you raise on how Greek or the particular setting of parliament speech is different from other settings concerning semantic change detection. We hope that our dataset can be a stepping stone for other researchers to answer these interesting questions. Specifically we mention:
>
> [lines 340-351]-> “​​Our dataset has a specific provenance, parliamentary recordings, and is not necessarily representative of language use and evolution in general. Yet, it can be useful in various applications of computational linguistics and political science, e.g., studies that examine whether word usage change behaves differently in different languages or contexts. Its extensive metadata can facilitate fine-grained semantic change studies, such as to evaluate whether a new parliament member gradually adjusts their speech to the style of the majority of speakers. It can be used for monitoring and tracking events and controversial topics over time [26, 21, 46, 10, 9] as well as rapid discourse changes during crisis events [45], or for classification of political texts [24]. Other applications can include political perspective detection [50] and viewpoint analysis [3] between parties, roles or genders and cross-perspective opinion mining [11, 41]. The dataset can be combined with datasets of tweets and public statements of parliament members, for modeling voting behavior and improving the tasks of roll call vote and entity stance prediction [37, 12, 49].”
>
> The importance of our work is also discussed in Section 1 “Introduction”:
>
> [lines 24-27] -> “To our knowledge, it is the only freely available dataset covering a comparable length of time in the Greek language. Moreover, by its nature as a record of the country’s parliament, it is again to our knowledge the only dataset that captures more than a quarter century of the recent Greek political history. ”
>
> [Response continues in the following comments]

---

> > ### Author Response · Authors · 2022-08-29
> > **Response - Part 2**
> >
> > We also mention in Section 2 “Related Work”:
> >
> > [lines 93-94] -> To the best of our knowledge, there are no existing studies on language change in modern Greek.
> > Concerning your question on “why we should prize variance over bias” we would like to note that our methodology aims to achieve the best of two worlds. On the one hand, in Section 4.1, we examine the stability of the approaches, in order to produce results that are reliable, that demonstrate lower variance and are not artifacts of the training parameters of the algorithms. On the other hand, we manually inspect the results for each approach, as presented in Section 4.2, in order to qualitatively evaluate whether they detect meaningful change.
> >
> > Furthermore, after your proposal to include confidence intervals, we have updated Fig. 3 (previously Fig. 2) accordingly. We also repeated the Compass experiments of Sections 5.2 and 5.3 50 times, each time introducing a different random seed and updated Fig. 3 (previously Fig. 2) and 4 (previously Fig. 3) accordingly, by computing the average semantic change for each word and including error bars, at the 95% confidence intervals calculated by the bootstrap method.  In some cases, the error bars are not visible because Compass demonstrates great stability of predictions.
> >
> > Concerning the presentation of the results of Fig. 3 (previously Fig. 2), we have greatly updated the discussion in order to better explain how the changes of semantic similarity in the graph are connected with real world events and to explain as many semantic changes as possible. After your helpful proposal, we also reference external data of refugee arrivals. We would like to clarify that in order to explain the semantic changes in the selected topics, we looked into their closest neighbors at each period in time. Our findings are not based on assumed correlations with historic events that happen to coincide with the usage changes. The updated discussion is presented below:
> >
> > [lines 281-297]-> “Fig. 4 shows a subset of topic embeddings that exhibit notable decrease in semantic similarity (≤ 0.65) over at least one pair of consecutive periods. After investigating the neighbors of the terms at each period, we conclude to the following findings. The word “macedonian” changes around 2000 from referring to a business consortium named “Macedonian metro”, to portraying the turbulence around the naming dispute between Greece and the Republic of North Macedonia. The semantic similarity drop between periods 14-15 and 16-17 corresponds to extensive debates that took place in the parliament nearing the dispute resolution in 2019. The word “refugee” changed from referring to cheap labor to reflecting the increased arrivals of refugees. The notable drop around 2015 is in agreement with external data recording an increased volume of refugee arrivals at the time. During the economic crisis, from 2008 onwards, “part-time” changed usage as it became a common employment practice, to reduce salary expenses. The terms “homosexuals” and “same-sex” emerge around 2007, reflecting an increasing social awareness. In the following years, the terms undergo important usage change as they approach the words “marriage”, “cohabitation agreement” and “adoption”. The word “man” changes context in 2019 from describing a male of the typical Greek family model or a criminal to referring to a policeman, associated with arbitrary police behavior and brutality. On the other hand, the word “woman” does not display notable usage change but we include it for comparison with the word “man”. “Woman” is constantly correlated with the words  “mother”, “child”, “spouse”, “family”, exhibiting a context limited to traditional family relations.”
> >
> > Concerning Fig. 5 (previously Fig. 4), the explanation provided for the semantic changes of political party embeddings is produced by looking into their closest neighbors at each period in time. To clarify this, we have complemented the presentation of the results with the following explanation:
> >
> > [lines 309-311] -> “We locate the period pairs for which each party embedding showcases the lowest semantic similarity and study its neighbors for each period to shed more detail to their usage change.”
> >
> > Finally, after your proposal to enrich the differences between methods with additional theoretical grounding, we have added the following sentences in Section 2 “Related Work”:
> >
> > [lines 42-44] -> “The former have shown to be mostly suitable for detecting changes of linguistic drift, more prominent in verbs, while the latter for capturing cultural semantic shifts, encountered more frequently in the nominal domain [27, 17]”
> >
> > [lines 59-60] -> “In their work, they propose that projection-based methods are more sensitive to proper nouns.”

---

### Official Review · Reviewer_bbUs · 2022-07-22
**Interesting dataset but analyses can be improved**

**Rating:** 6
**Confidence:** 4
**Clarity:** The paper is generally well written.

**Strengths:**

- The process to create the dataset is very well documented.
- The paper is generally clearly written.
- Potentially useful resource due to its characteristics: 1989-2020, Greek (non-English, which I think is nice and complements existing English datasets), large, rich metadata, parliamentary record files (interesting for political analysis), data from multiple sources.


**Weaknesses:**

- The analyses in the paper are too shallow, leaving open many questions about both the dataset and the phenomena the authors intend to study; I suggest focusing on only one or two analyses and executing them well.  Furthermore, they do not demonstrate the full potential of the dataset. For example, semantic change is analysed by comparing between two periods, rather than doing, say, a year by year analysis–I wonder if that means their data is actually not large enough to do more fine-grained analyses. I also wished they had made more use of their rich metadata, such as speaker information.
- The analyses are not well described (some steps are unclear) and some claims are not well supported.


**Additional Feedback:**

- Why remove punctuation except full stops? (161)
- How many speakers are included in total?
- Table 4: “Words with significant usage change” Was some test performed to determine significance?
- Although the dataset covers a large time period, semantic change is done by comparing two decades (228). So I wonder if this dataset is suitable to detect change on a finer level (e.g. year).
- Figure 2 is unreadable in black and white.
- One of the contributions of this paper is to evaluate algorithms for detecting semantic shift. However, I think there are several issues with this: 1) The literature review does not discuss methods to evaluate semantic change detection approaches; 2) They only focus on stability and ignore other quality aspects of semantic change detection approaches; 3) It’s not clear why they want to do stability analysis. Is this to reproduce results from previous literature? Do they think that the characteristics of this dataset warrant new stability investigations? I actually feel that this part doesn’t fit well into the paper, so I would suggest removing the stability analysis and focusing on the other parts of the paper.


**Correctness:**

- **"significant"**: This paper is using ‘significant’ many times (e.g. line 201) while they do not seem to perform significant tests. I prefer not using ‘significant’ unless some statistical testing was involved.
- **Gender assignment**: This paper assigns gender to speakers based on popular female & male names. Given that this dataset is potentially interesting for its gender dynamics (e.g. in parliament) I think it’s important that gender is assigned correctly. I’m not very familiar with Greek, so I don’t know to what extent names signal gender and whether there are names that are not exclusively used for a certain gender. What is the accuracy of this method? Is it possible to manually correct/assign gender to speakers instead of using a heuristic based on names? (how many speakers are there?). Were you able to assign gender to all speakers based on this approach?
- **Evaluation of semantic change detection approaches**: The paper aims to compare different semantic change methods, however the analysis is not systematic enough. Certain statements lack justification, for example line 200 “The best performers were Compass and Orthogonal Procrustes. A significant portion of their top 100 most changed words were proper nouns and names of politicians.” Why are these the best performers? (just based on the proportion of proper nouns & names?) Furthermore, instead of saying “a significant portion” it would be useful to quantify the exact proportions. Furthermore, they show a  “representative selection of results” (Table 3/line 201): It would be informative to just show the top x changed words, rather than doing a manual selection. How do we know that these words are indeed representative? Generally, a more systematic evaluation would be informative. For example, inspect the top 100 words of each method and annotate whether they seem to reflect real semantic change. Fig 3: “Fig. 3 shows a subset of topics whose embeddings exhibit significant” Can you be more explicit regarding how this selection was done? When was the change considered to be “significant” How many topics were included/excluded?


**Documentation:**

It would be good to see an explicit note regarding the license of this dataset, and generally whether the crawling/sharing of the data is allowed.

**Ethics:**

I don't believe so, since these are parliamentary records so I assume this is public data.

The only possible concern is gender assignment (see comment above).

**Relation To Prior Work:**

- The related work focuses on semantic change detection methods (calling these “state of the art”, line 32), but it only covers approaches using static word embeddings and not more recent approaches using contextual embeddings (e.g., BERT). I think it’s actually fine for this paper to just use static embedding approaches, but the paper needs to acknowledge more recent approaches.
- The paper aims to contribute to the evaluation of semantic change approaches but doesn’t discuss evaluation approaches (e.g. https://aclanthology.org/D19-1007/) or what criteria such an evaluation should focus on. They only discuss “stability”, but are missing relevant papers in this area on embedding stability: https://aclanthology.org/Q18-1008/ and https://aclanthology.org/2021.emnlp-main.476/.

**Summary And Contributions:**

Thanks for replying in such detail and making the changes.
Small comment: could you clarify in the paper how you arrived at 0.65 (seems a bit arbitrary).
I've increased my score slightly based on their response.

====
This paper presents a dataset of Greek Parliament Proceedings covering 1989 to 2020. The temporal component makes it an interesting resource for studies in NLP and political analysis. The paper describes the creation of the dataset and presents several analyses focusing on 1) stability of semantic change detection models 2) temporal trends in the data related to language change.

---

> ### Author Response · Authors · 2022-08-29
> **Response - Part 1**
>
> Thank you for your kind comments.
>
> As per your proposal, we have devoted an additional paragraph in Section “Related Work” to discuss more recent approaches using contextual embeddings. Specifically we mention:
>
> [lines 67-74]->“Furthermore, the rise of contextual embeddings such as BERT [6] and ELMo [40] has enabled  important developments in the study of word usage change as they are capable of generating a different vector representation for each specific word usage. Contextual embeddings can be used  in the context of usage change detection by aggregating the information from the set of token embeddings [36, 30, 31, 25, 15]. However, related work shows that, for the time being, it is complex to disambiguate between word senses, and there is a large disparity between results on different corpora [31, 30, 36, 28]. Finally, recent studies have emerged that ensemble multiple types of word embeddings and distance metrics to experiment on improving overall performance [25, 32].”
>
> We have also added the following statement in the same section:
>
> [lines 96-99]->“The selection of the approaches for language change detection aims to be representative of different established methodologies proposed in the related work and does not constitute a complete benchmark evaluation on language change detection methods. ”
>
> Concerning your interest in a year-by-year analysis of the dataset for the detection of semantic shifts, as proposed in the related work, semantic shifts are a process that requires corpora that are separated by longer periods of time [27]. Semantic change of words is not usually studied in such short periods as consecutive years. Furthermore, our decision to study the usage change between parliamentary periods or decades is based on the need to identify changes in the political environment, which usually unfold in longer periods of time. Cultural shifts in word usage might take place in shorter periods of time due to a force majeure such as the economic crisis or a pandemic. Our dataset is large enough to perform such a study for all years except 1989 and 2020, for which our data do not span 12 months, as well as the year 1995, for which no parliamentary records have been published on the website of the Greek parliament, as mentioned in the manuscript. Also, and perhaps most important, please note that the constitution of the parliament remains essentially the same between elections (some members of parliament may be indicted, or die, but such cases are rare), so the composition of the speaker corpus does not change on a year-by-year basis, but on a parliamentary period by parliamentary period basis, which is the one we adopted here.
>
> [Response continues in the following comments]

---

> > ### Author Response · Authors · 2022-08-29
> > **Response - Part 2**
> >
> > Regarding your proposal to facilitate additional metadata of the dataset for analysis, we added Fig. 1 in Section 3.1, which depicts the percentage of female members per parliamentary period and per political party, the total percentage of female members of the parliament per parliamentary period as well as the percentage of characters of speech
> > delivered by female individuals, again per period and per political party. We included the following discussion of the results:
> >
> > [lines 109-131] -> “Delving deeper into our dataset, Fig. 1 depicts the percentage of female members in the Greek Parliament and the percentage of characters of speech delivered by female individuals, per political party and per parliamentary period. The difference between the membership percentage and the speech percentage is highlighted with dotted vertical lines. For reasons of readability, we depict political parties that have played an important role in recent political history. These are New Democracy (center-right, hereafter ND), the Panhellenic Socialist Movement (center-left, hereafter PASOK), the Coalition of the Radical Left—Progressive Alliance (left, hereafter SYRIZA), the Communist Party of Greece (communist, hereafter KKE), the Coalition of the Left, of Movements and Ecology (left, hereafter SYN) and Golden Dawn (extreme right, nationalist, nazi-fascist, hereafter GD). We exclude period 14, which lasted two days and was a transitional government.
> >
> > Concerning membership, the total percentage of female individuals (dashed line) increases over time. Left-wing political parties like SYN, KKE and SYRIZA achieve higher percentages of female members and remain above the total percentage of female members for almost all periods. The percentage of the center-left political party of PASOK presents great fluctuation over the years. On the other hand, the percentage of the center-right political party of ND remains below the total average percentage. Lastly, the far-right political party of GD has the lowest percentage of female members compared to the selected political parties. Regarding the participation of females in parliamentary debates, only the left-wing political parties of SYN and KKE achieve percentages higher than that of female membership for most periods. Overall, none of the examined parties has a percentage of female members equal to or greater than 50% at any point in time. After investigating the rest of the parties, we found that only two left-wing parties have achieved percentages greater than 50% for female members, namely Alternative Ecologists (greens, left, 100% for periods 6 & 7) and MeRA25 (left, 55% for periods 16 & 18).”
> >
> > In order to resolve any valid concerns on the statistical significance of our results, we repeated our experiments for the computations of usage change of selected topics and political party names 50 times each, with 50 different random seeds. We produced error bars at the 95% confidence intervals calculated by the bootstrap method and included them in the respective figures, namely Fig. 4 (previously Fig. 3) and Fig. 5 (previously Fig. 4). In most cases, the error bars are not visible, because Compass computes usage change with stability. We have also replaced the word significant with synonyms, especially in Section 5,  where it was most used. Finally, we included error bars in Fig. 3 (previously Fig. 2) that studies the stability of each approach for the detection of word usage change.
> >
> > [Response continues in the following comments]

---

> > > ### Author Response · Authors · 2022-08-29
> > > **Response - Part 3**
> > >
> > > Regarding your concerns about the possibility of misgendering the members of the parliament, Greek names have genders. Some very few unisex names are transfers from other languages: for instance, the names “Vivian / Vivianne” in Greek would be transferred as “Βίβιαν”, with no distinction between male and female. It is only very recently that activists have started using neutral names (which traditionally do not exist in Greek) or use names of the opposite gender. No person with a name that cannot be exactly attributed to its gender has been elected to the Greek parliament yet. In addition, during the creation of the dataset, we followed a list of steps to ensure the best accuracy of our results, since our process was based on external data resources. In the script “add_gender_to_members.py”, we compute the intersection of female and male names from the lists of Wikipedia entries in order to define any cases of unisex Greek names, which is not common in the Greek language. If the name of a parliament member consists of any unisex names, the script outputs a message to the user in order to manually inspect the gender assignment. In our case, no unisex names were found in the names of the parliament members. Furthermore, in case any parliament member name is not found in the Wikipedia entries, the script again outputs it to the user. With the use of this safety measure, we detected 49 names that could not be found in the Wikipedia entries. We manually assigned the gender to these names, by extending the female and male name lists from Wikipedia accordingly, as it can be seen in the source code of the script. Finally, when more than one first name and/or an additional nickname are provided for a member of the parliament, we manually deduced the gender of all the names and checked if the results are in agreement. This led to making a manual correction of a single entry that referred to a female member but one of her names was actually officially male. That is the case of “Πουλου Λεωνιδα Παναγιου (Γιωτα)”. In this string the order of the names is “Last_name Father’s_name First_name (Nickname)”.
> > >
> > > In order to address any further doubts concerning the validity of the gender assignment, we manually inspected all the assigned genders in the file “all_members_activity.csv”. This file includes 4939 entries that correspond to 1666 unique names of members of parliament. From our inspection, we found only one mistaken instance out of the 1666 unique names, spanning in two out of the 4939 entries/lines. This mistake was not transferred to the main dataset file tell_all_cleaned.csv. It was caused by an incorrect conversion of the member's name from the genitive to nominative case and concerns data collected from the website of the Secretariat General for Legal and Parliamentary Affairs (https://gslegal.gov.gr), as explained in Section 3.3, paragraph "Government Members" of the manuscript. We corrected the file “all_members_activity.csv” accordingly in our Zenodo upload.
> > >
> > > Furthermore, the total number of unique identified speakers in the “tell_all_cleaned.csv” file is 1523. The total number of parliament and government members from 1989 up to 2020 contained in the “all_members_activity.csv” file is 1666.
> > >
> > > Concerning any questions raised by Fig. 4 (previously Fig. 3), we depict topics whose cosine similarity reached a value lower or equal to 0.65. We have added this detail in the revised manuscript, specifically stating:
> > >
> > > [lines 281-282] -> “Fig.4 shows a subset of topic embeddings that exhibit notable decrease in semantic similarity (≤0.65) over at least one pair of consecutive periods”.
> > >
> > > We also changed the description of Fig. 3 (previously Fig. 2) to mention the exact number of the depicted topics. The figure description now says: “Usage change of 10 political topics between pairs of parliamentary periods”.
> > >
> > > Furthermore, the supplementary material included a detailed list of all the studied topics. We complemented the list with a specific mention of the total number of topics. We also added this information in the revised manuscript. Specifically, we mention:
> > >
> > > [lines 275-278] -> “For the selection of topics, we consulted the website of Vouliwatch, a non-partisan parliamentary monitoring organization that provides an extensive comparison of party positions on 69 topics of significant political interest. We extended this list with 22 additional topics, selected for their popularity.”
> > >
> > > In addition, we have changed the description of Table 3 from “A selection of words with significant usage change between the 1990s and 2010s per method.” to “A representative selection of words from the top 100 most changed words per approach between the 1990s and 2010s.”, following your comments.
> > >
> > > [Response continues in the following comments]

---

> > > > ### Author Response · Authors · 2022-08-29
> > > > **Response - Part 4**
> > > >
> > > > Regarding your concerns on the qualitative evaluation of the methods, you propose that we  manually inspect the top 100 words of each method and evaluate whether they seem to reflect real semantic change. Fortunately, we did that and already included a file in our Zenodo upload, namely top100_minfreq50.xls. It includes the top 100 words of each method which we manually evaluated. As mentioned in Section 5.1.6 “Stability results”, we created one csv file per approach with the top 100 most changed words. All the files were also merged in the file top100_minfreq50.xls for the reviewers’ convenience. We manually inspected the results and chose to include the most representative, due to the paper size limitations of the conference. You also propose that instead of a representative selection, we could include in the paper the top x changed words per approach. However, we would need a large number of top results in order to gather representative cases of each approach. Unfortunately, the top 5 or top 10 results of each approach are not representative of the approach. Furthermore, we have removed from the manuscript the following text: “The best performers were Compass and Orthogonal Procrustes. A notable portion of their top 100 most changed words were proper nouns and names of politicians”. We understand why these statements can be confusing and they did not convey the information we wanted. In the second and third paragraphs of Section 4.2 “Qualitative Evaluation: Top Changed Words between 1990-1999 & 2010-2019”, we go into detail on why NN and Second-Order Similarity approaches provided less explainable results, while Compass and Orthogonal Procrustes detected explainable meaningful usage changes. Finally, we rephrased the statement “As Compass presented the best performance in terms of stability and detection of meaningful change, we used it to investigate changes in word usage and events in Greece's recent political history” to “As Compass presented the best performance combination in both stability and detection of meaningful change, we used it to investigate changes in word usage and events in Greece's recent political history” [lines 251-252].
> > > >
> > > > Regarding your questions about why we chose to remove all punctuation except full stops, removal of punctuation marks is a common practice in data preprocessing, also applied in related work [18]. Most importantly, full stops are useful for tokenizing text into sentences. The Compass tool uses gensim.models.word2vec.LineSentence() to iterate over the training corpus. The LineSentence() method takes as input a file where each sentence is written in a newline.
> > > >
> > > > Furthermore, you mention that our paper “aims to contribute to the evaluation of semantic change approaches but doesn’t discuss evaluation approaches”. We would like to clarify that the aim of our paper is to present a new dataset in a lean-resource language and in the context of political corpora. Our purpose is mainly to showcase the value of the dataset in the context of language change study. For this reason we have enriched Section 6 “Conclusions and Future Work” with applications for which our dataset can prove useful. We mention:
> > > >
> > > > [lines 340-351]-> “​​Our dataset has a specific provenance, parliamentary recordings, and is not necessarily representative of language use and evolution in general. Yet, it can be useful in various applications of computational linguistics and political science, e.g., studies that examine whether word usage change behaves differently in different languages or contexts. Its extensive metadata can facilitate fine-grained semantic change studies, such as to evaluate whether a new parliament member gradually adjusts their speech to the style of the majority of speakers. It can be used for monitoring and tracking events and controversial topics over time [26, 21, 46, 10, 9] as well as rapid discourse changes during crisis events [45], or for classification of political texts [24]. Other applications can include political perspective detection [50] and viewpoint analysis [3] between parties, roles or genders and cross-perspective opinion mining [11, 41]. The dataset can be combined with datasets of tweets and public statements of parliament members, for modeling voting behavior and improving the tasks of roll call vote and entity stance prediction [37, 12, 49].”
> > > >
> > > > [Response continues in the following comments]

---

> > > > > ### Author Response · Authors · 2022-08-29
> > > > > **Response - Part 5**
> > > > >
> > > > > We would also like to note that the evaluation process as well as the selection of approaches for language change detection aims to be representative of different established methodologies proposed in the related work. It does not constitute a complete benchmark evaluation on language change detection methods but aims at highlighting the utility of our dataset. For this reason, we added the following statement in Section 2 “Related Work”:
> > > > >
> > > > > [lines 96-99]->“The selection of the approaches for language change detection aims to be representative of different established methodologies proposed in the related work and does not constitute a complete benchmark evaluation on language change detection methods. ”
> > > > >
> > > > > Concerning the questions raised on the aspect of stability, we have enriched the discussion in Section “Related Work” with the proposed and highly relative papers you mention [44, 2, 4]. We have also complemented the paragraph introducing the concept of stability with an explanation of the importance of this aspect as well as the reason we measure it in our study. Specifically we mention:
> > > > >
> > > > > [lines 75-85] ->“Different approaches can give different results, thus comparing them is a challenge [44]. An additional challenge is the stability of the approach used. An approach demonstrates stability when slightly different runs on a dataset do not significantly affect the results [16]. Recent studies highlight the importance of stability, as a high variation can be a sufficient reason to call the whole method into question [2, 4]. Researchers have identified a number of factors that affect stability, including properties of the underlying algorithms used to construct the embeddings [16, 47, 2, 29, 20]. Subsequent runs of word embedding algorithms on the same data will not necessarily produce the same results, due to the stochastic nature of the approaches. Gonen et al. [16] use intersection@k, mentioned above, to gauge the stability between the predictions of two different runs of the same algorithm. We adopted this metric in our work, in order to select a stable usage change algorithm for our study.”
> > > > >
> > > > > Finally, we have corrected Fig. 3 (previously Fig. 2) and all other figures of the paper, so that they are readable in black and white. Thank you for pointing it out!

---

### Official Review · Reviewer_vnmy · 2022-07-27
**Recommending accept after author response**

**Rating:** 7
**Confidence:** 4
**Correctness:** There are no correctness issues.

**Strengths:**

- presents an organized dataset from unorganized sources
- provides valuable resources to the low-resource Greek language
- the proposed dataset has implications for both natural language processing and computational political science

**Weaknesses:**

I hope the following suggestions would make the paper stronger:

- I appreciate the authors' work to evaluate four approaches on the new dataset. There are some more up-to-date works on language change detection such as [1,2]. I hope the authors could discuss them in the related work and even evaluate them as well.

[1] Montariol, Syrielle, Matej Martinc, and Lidia Pivovarova. "Scalable and interpretable semantic change detection." Proceedings of the 2021 Conference of the North American Chapter of the Association for Computational Linguistics Human Language Technologies. The Association for Computational Linguistics, 2021.

[2] Liu, Yang, Alan Medlar, and Dorota Glowacka. "Statistically Significant Detection of Semantic Shifts using Contextual Word Embeddings." The 2nd Workshop on Evaluation & Comparison of NLP Systems. The Association for Computational Linguistics, 2021.

- The authors did a great job in presenting every single detail of dataset collection and experiment settings. However, a major concern of mine is that the main paper (especially in Sections 3 & 4) is also filled with wordy descriptions of trivial details. For example, the name of each individual file (line 94), the value of random seeds (line186), and many other trivialities like these could be moved to the supplementary material. This would make space for more discussion of the new dataset's implications in computational linguistics and political science, which, in my opinion, should be the focus of this paper.

- What is intersection@k, specifically? Do better methods have higher or lower intersection@k? Since Figure 2 is the only quantitative experiment in the paper, I feel that more details could be provided to help non-expert readers better understand the results.

- In the paper checklist 3(c), the authors stated that they reported error bars of experiments. Maybe I missed something, but I did not come across the error bars of results in Figure 2. This is especially important since "Compass with frequency cutoffs" and "NN" seem to have near-identical performance when k is large.

- I personally feel that the references in this paper are somewhat limited. Experiments in Sections 4 and 5 are related to many works in the intersection of NLP and computational political science, and I hope the authors could better position their work in the context of prior computational political science works such as [3,4,5,6].

[3] Zhang, Wenqian, et al. "KCD: Knowledge Walks and Textual Cues Enhanced Political Perspective Detection in News Media." In NAACL 2022.

[4] Mou, Xinyi, et al. "Align Voting Behavior with Public Statements for Legislator Representation Learning." In ACL 2021.

[5] Feng, Shangbin, et al. "Legislator Representation Learning with Social Context and Expert Knowledge." In Arxiv.

[6] Yang, Yuqiao, et al. "Joint representation learning of legislator and legislation for roll call prediction." In IJCAI 2020.





**Additional Feedback:**

Please see above.

**Clarity:**

The paper is generally well-written. However, it would be better if the authors move trivial details to the appendix and expand on experiments and discussions.

**Documentation:**

The authors did a great job in providing comprehensive documentation.

**Ethics:**

There are no ethical concerns.

**Relation To Prior Work:**

Some prior works in semantic change detection and computational political science are missing. I hope the authors would discuss previous works that study legislator speeches, public statements, and political text to better position their work.

**Summary And Contributions:**

This work presents a large-scale greek parliament proceedings dataset ranging from 1989 up to 2020. In addition to the speech content, well-organized metadata is also provided in the dataset. The authors explored its potential in computational linguistics and computational political science research. Detailed documentation of everything related to data collection and processing is also provided.

---

> ### Author Response · Authors · 2022-08-29
> **Response**
>
> Thank you for your constructive comments.
>
> The related work you mentioned is highly relative to our work and addressing your comment has definitely improved the quality of our study. We enriched the related work with a discussion of all the papers you proposed, along with the opportunities and limitations they present. Specifically we mention:
>
> [lines 67-74]->“Furthermore, the rise of contextual embeddings such as BERT [6] and ELMo [40] has enabled  important developments in the study of word usage change as they are capable of generating a different vector representation for each specific word usage. Contextual embeddings can be used  in the context of usage change detection by aggregating the information from the set of token embeddings [36, 30, 31, 25, 15]. However, related work shows that, for the time being, it is complex to disambiguate between word senses, and there is a large disparity between results on different corpora [31, 30, 36, 28]. Finally, recent studies have emerged that ensemble multiple types of word embeddings and distance metrics to experiment on improving overall performance [25, 32].”
>
> We would also like to note that our selection of approaches for language change detection aims to be representative of different established methodologies proposed in the related work. It does not constitute a complete benchmark evaluation on language change detection methods but rather aims at highlighting the utility of our dataset. We added the following statement in Section 2 “Related Work” [lines 96-99].
>
> As mentioned in the work of Yang et. al. [30], “existing semantic change detection methods are problematic to apply if (i) the data set is of limited size or (ii) you need to estimate the semantic shift for words with low term frequencies”. In our study, we do our best to eliminate these contingencies by studying word usage change between corpora of large size, with significant vocabulary overlap and we eliminate from our study words with very low term frequencies (see Sections 3.5, 4.1, 4.2). Finally, we would be really interested in studying and extensively comparing the approaches that facilitate clustering and contextual embeddings with neighbor-based or projection-based approaches. However, in order to implement such a study properly, we feel this should be the topic of a whole new paper. Fortunately, our dataset can facilitate such a study for the Greek language and we would certainly be interested in doing so in the future.
>
> As per your proposal, we moved to the supplementary material implementation details from Sections 3 and 4, concerning the record collection, the creation of the support datasets, the speech extraction, preprocessing and the implementation of the stability study.
>
> In addition, we rephrased the section that introduces the intersection@k metric and discusses Fig. 3. Please refer to lines 209-226 of the revised manuscript.
>
> Fig. 3 shows the average intersection@k between the 45 pairs of different runs for each approach, with 95% confidence intervals calculated by the bootstrap method. The NN method exhibits the greatest stability even for very small values of k. This means that for all values of k and regardless of the random seed used, the changed words this approach detected were mostly the same. Compass follows closely, while the Compass variation yields better stability results, similar to that of NN. The Orthogonal Procrustes and the Second-Order Similarity approach gave worse stability results, with the latter even decreasing for k > 100.”
>
> Furthermore, higher stability in a method means that the predictions are not artifacts of the selection of a random seed or other parameters of the implementation.
>
> As you also proposed, we further discuss our dataset's implications in computational linguistics and political science. We have added an extra paragraph in Section 6 “Conclusions and Future Work”, elaborating on the usability of our dataset on specific tasks and complemented our paper with 14 new citations. Please refer to lines 340-351of the revised manuscript.
>
> Finally, we have corrected Fig. 2, 3 and 4 which are now numbered as 3, 4 and 5. The figures now include error bars, at the 95% confidence intervals, calculated by the bootstrap method, in order to support our claim of paper checklist 3(c). For Fig. 4 and 5, we repeated our experiments 50 times, each time introducing a different random seed. Thank you for thoroughly evaluating our submission and pointing it out.

---

> > ### Comment · Reviewer_vnmy · 2022-08-29
> > **Thank you for your response.**
> >
> > My concerns are adequately addressed and I am raising my score to reflect that. Thank you for your efforts!

---

### Official Review · Reviewer_x48T · 2022-07-27
**Review on Submission 185**

**Rating:** 6
**Confidence:** 3
**Correctness:** The data construction is sounding.
**Clarity:** The paper is clear and well written.

**Strengths:**

(1) The code and documentation were well-organized for accessibility and reproducibility.

(2) The dataset provided a large-scale political speech dataset for 30 years with rich metadata, which can be useful for future applications.

(3) The authors provided applicable scenarios that the dataset can be used for computational analysis in political science and linguistics in Section 4.


**Weaknesses:**

(1) In Section 5.3., the paper presented the usage change of political part names by analyzing cosine similarities between the parliamentary periods. The authors provided the details on how to measure the cosine similarities in Section 6.3. in the Supplementary materials. However, the information on how to measure the usage change of political party name embeddings in the manuscript was limited. The authors should briefly explain their method in the manuscript.

**Additional Feedback:**

N/A

**Documentation:**

The sufficient details on data collection and preprocessing were provided with the codes. The documentation was well-organized.

**Ethics:**

There is no ethical concern.

**Relation To Prior Work:**

The authors argue that the main difference is that the proposed dataset focused on political speeches in the low-resource language, Greek, compared to the prior efforts. Also, the authors reviewed and applied various language change detection approaches to the proposed dataset in Section 4.1.

**Summary And Contributions:**

This paper introduced a dataset of the Greek parliament proceedings, which included 1,280,918 speeches of parliament members from 5,355 parliamentary sitting record files with sufficient metadata. By applying the word usage change detection models and the proposed dataset, the authors analyzed word usage change in the context of Greek political environments.

This dataset contributes to capturing diachronic semantic shifts of words for low-resource languages such as Greek. In addition, this dataset provides a rich context of political environments in Greek, which can be useful in a political science domain.

---

> ### Author Response · Authors · 2022-08-29
> **Response**
>
> Thank you for your time and kind comments.
>
> Concerning Section 5.3 “Usage Change of Political Party Name Embeddings”, we enriched the information provided with a further explanation of the computation process, by adding the following:
>
> [lines 300-305] -> “As mentioned in Section 3.4, we replaced all political party references with the symbol “@" followed by an abbreviation of the party name, using regular expressions that capture grammatical cases and variations.  We trained Compass between consecutive pairs of parliamentary periods and computed the cosine distance between the vectors of political party names. We repeated the computations with 50 different  random seeds and produced error bars at the 95% confidence intervals with the bootstrap method.”
>
> Finally, we have enriched Section 6 “Conclusions and Future Work”, in order to better demonstrate the usability and importance of our dataset, as follows:
>
> [lines 340-351]-> “​​Our dataset has a specific provenance, parliamentary recordings, and is not necessarily representative of language use and evolution in general. Yet, it can be useful in various applications of computational linguistics and political science, e.g., studies that examine whether word usage change behaves differently in different languages or contexts. Its extensive metadata can facilitate fine-grained semantic change studies, such as to evaluate whether a new parliament member gradually adjusts their speech to the style of the majority of speakers. It can be used for monitoring and tracking events and controversial topics over time [26, 21, 46, 10, 9] as well as rapid discourse changes during crisis events [45], or for classification of political texts [24]. Other applications can include political perspective detection [50] and viewpoint analysis [3] between parties, roles or genders and cross-perspective opinion mining [11, 41]. The dataset can be combined with datasets of tweets and public statements of parliament members, for modeling voting behavior and improving the tasks of roll call vote and entity stance prediction [37, 12, 49].”
>
> In general, we have addressed the comments of all the respectful reviewers, which we believe have made our paper stronger.

---

### Official Review · Reviewer_2qgz · 2022-07-28
**Good Dataset and Interesting Experiments**

**Rating:** 7
**Confidence:** 3
**Correctness:** The dataset and experiments seem sound.

**Strengths:**

1. Sizeable dataset (both in number of documents and timespan) in language where data is less available. As the authors state, it can be useful both for computational linguistics/NLP applications and for political science ones, and combinations thereof.

2. The experiments on changes in language usage over time, while perhaps not as extensive as e.g. a method-focused paper, nonetheless produce a number of meaningful/interpretable results. They may be interesting of their own right for future researchers, and help demonstrate the value of this dataset.

3. Documentation for usage and reproducibility is generally thorough.



**Weaknesses:**

1. When doing the entity resolution with string similarity, the process and 0.95 threshold seem reasonable, but it would be better if there were post-hoc/quantitative validation. For example, one could take 50 or 100 random names that were matched and the same number that were close to the threshold but just below, and manually label if the former are indeed correct matches and the latter correctly left unmatched.



**Additional Feedback:**

Small suggestion for future work: it would be great if this data were updated in the future with the past couple years, since there may be interesting shifts in political discussions due to Covid.

Thank you for the solid dataset and experiments.

**Clarity:**

The paper is well written.

I noticed two typos:

- line 140 "For speeches delivered holders" - missing "by" ("by holders")

- line 81 in appendix "donwload" -> "download"

**Documentation:**

The dataset and experiments are well-documented, both in the main paper and extensive details in the appendix.

The dataset hosting/maintenance is clear and available, but I didn't see a license for it (might have missed it somewhere).

Line 156-157 "For matching we used yet another dataset of 475 names and nicknames, which cannot be shared due to licensing reasons." - fair enough that it can't be shared, but what is it/where does it come from?

**Ethics:**

The dataset includes a gender variable. Is there any potential of misgendering individuals in the data, either through errors or in case their gender does not match the options? The paper's approach to and usage of the gender variable do seem reasonable. But I'm not very familiar with the mechanics of Greek names or demographics of Greek politicians, so would welcome any thoughts from the authors on this.

I don't have any other ethical concerns.



**Relation To Prior Work:**

The relation to prior work seems clear. There are some related datasets for other languages and countries, but no comparable Greek dataset. The paper also has a clear discussion approaches to diachronic semantic shifts.

**Summary And Contributions:**

The paper provides a dataset of Greek parliament speeches. It also experiments on changes over time in word usage in those speeches.

---

> ### Author Response · Authors · 2022-08-29
> **Response**
>
> Thank you for reading our paper and for your kind comments.
>
> Regarding your question about the existence of a manual validation of the 0.95 string similarity threshold, we added a manual evaluation of a sample of 150 unique entity pairs, as depicted in Table 2, in Section 4.6 of the supplementary material. The table is accompanied by a paragraph explaining the evaluation process and the results [lines 214-227].
>
> Concerning the licensing of the dataset, the Zenodo platform provides a specified field for users to define the license that accompanies their upload. The platform displays the license on the right side of the upload page. We have selected the Creative Commons Attribution 4.0 International license for all uploads. In order to make this information more accessible, we specifically mention the license in Section 7 “Dataset License & Maintenance” of the supplementary material:
>
>  [line 445]-> “..    .under the Creative Commons Attribution 4.0 International license …”
>
> Concerning your question about the origin of the additional dataset of 475 names and nicknames, it was collected from a website of Greek name-day information, namely https://www.onomatologio.gr. The terms of use of the website do not allow the public redistribution of its content but allow a local copy of its contents’ on a personal computer.
>
> Regarding your concerns about the possibility of misgendering the members of the parliament, Greek names have genders. Some very few unisex names are transfers from other languages: for instance, the names “Vivian / Vivianne” in Greek would be transferred as “Βίβιαν”, with no distinction between male and female. It is only very recently that activists have started using neutral names (which traditionally do not exist in Greek) or use names of the opposite gender. No person with a name that cannot be exactly attributed to its gender has been elected to the Greek parliament yet. In addition, during the creation of the dataset, we followed a list of steps to ensure the best accuracy of our results, since our process was based on external data resources. In the script “add_gender_to_members.py”, we compute the intersection of female and male names from the lists of Wikipedia entries in order to define any cases of unisex Greek names, which is not common in the Greek language. If the name of a parliament member consists of any unisex names, the script outputs a message to the user in order to manually inspect the gender assignment. In our case, no unisex names were found in the names of the parliament members. Furthermore, in case any parliament member name is not found in the Wikipedia entries, the script again outputs it to the user. With the use of this safety measure, we detected 49 names that could not be found in the Wikipedia entries. We manually assigned the gender to these names, by extending the female and male name lists from Wikipedia accordingly, as it can be seen in the source code of the script. Finally, when more than one first name and/or an additional nickname are provided for a member of the parliament, we manually deduced the gender of all the names and checked if the results are in agreement. This led to making a manual correction of a single entry that referred to a female member but one of her names was actually officially male. That is the case of “Πουλου Λεωνιδα Παναγιου (Γιωτα)”. In this string the order of the names is “Last_name Father’s_name First_name (Nickname)” and the First_name Παναγιου is male.
>
> In order to address any further doubts concerning the validity of the gender assignment, we manually inspected all the assigned genders in the file “all_members_activity.csv”. This file includes 4939 entries that correspond to 1666 unique names of members of parliament. From our inspection, we found only one mistaken instance out of the 1666 unique names, spanning in two out of the 4939 entries/lines. This mistake was not transferred to the main dataset file tell_all_cleaned.csv. It was caused by an incorrect conversion of the member's name from the genitive to nominative case and concerns data collected from the website of the Secretariat General for Legal and Parliamentary Affairs (https://gslegal.gov.gr), as explained in Section 3.3, paragraph "Government Members" of the manuscript. We corrected the file “all_members_activity.csv” accordingly in our Zenodo upload.
>
> Finally, we would be really excited to extend the dataset to include the past couple of years and study the effect the pandemic had in the topics discussed in the parliament as well as any semantic shifts caused in the use of words. There has already been interest from researchers from the field of political sciences in the dataset and since all the scripts are documented, up-to-date and running smoothly, it would be easy for us or any other researcher to repeat the process for any new data points.

---

### Official Review · Reviewer_9C2V · 2022-07-28
**Reproducible Greek Parliament Proceedings Dataset for Computational Linguistics and Political Analysis**

**Rating:** 7
**Confidence:** 3

**Strengths:**

Strengths
Describe the strengths of the dataset and/or benchmark. Typical criteria include: significance of the contribution, relevance to the broader research community, accessibility and accountability, and ethical and social implications.

===

Overall good and interesting paper, insightful.

More importantly however contribution is relevant for research community due to broad availability of parliamentary proceedings in multiple countries and it's providing blueprint for creating similar datasets and examplatory analysis. It can be easily propagated to other low resource languages.

Moreover authors ensured that source and results of paper are broadly and easily accessible and thanks to following reproducible research principles ensured processes can be traced and results are judged as highly accountable.

Reviewer did not found profound ethical or social implications.


**Weaknesses:**

Weaknesses
Explain the limitations of this work along the same axes as above.

===

There are no major weaknesses of a paper.
There are two minor things that could be addressed.
Parsing was performed with Apache Tika and there is no mention about evaluating alternatives: PyPDF2, Textract, pdfPlumber, pdfMiner3 to name a few. Differences in parsing quality can be substantial and have effect on experiment results.
Moreover what could be evaluated and discussed is setting a cutoff of word occurrences to other values than 200. It’s interesting how different values of this parameter could influence analysis and results, especially that many other parameters were evaluated and this one, which has quite high potential for influencing experiment quality. There is explanation of the choice but it could justify as well maybe any other value.


**Additional Feedback:**

Additional Feedback
Please provide any additional feedback, comments, suggestions for improvement and questions for the authors

===

On https://zenodo.org/record/6626316#.YuBjKITP3Vh in spite of downloading dataset, value of downloads counter still shows 0 - authors should solve this as it has negative impact on perception.


**Clarity:**

Clarity
Is the paper well written?

===

Yes, paper is well written with clarity and has a good flow. Numerous references, informative graphics and context were provided where needed, what made article easy to follow.

**Correctness:**

Correctness
Are the claims made in the submission correct? If the submission is a dataset, it is constructed in a sound way? If it is a benchmark, are the evaluation methods and experiment design appropriate and performed correctly?

===

Yes, submission is correct and orderly structured. Dataset is solid, especially decisions that were made along the way are well explained, what is unfortunately rare. There’s not much more to crawl in domain that paper discusses, so it’s also complete. Authors attention to detail is demonstrated in multiple places.

**Documentation:**

Documentation
For datasets, is there sufficient detail on data collection and organization, availability and maintenance, and ethical and responsible use? Note that dataset submissions should include documentation and intended uses; a URL for reviewer access to the dataset; and a hosting, licensing and maintenance plan. For benchmarks, is there sufficient detail to support reproducibility?

===

Documentation is exhaustive, well written and informative. Repository and code structure are well thought and clear repro steps are provided to ensure reproducibility.

**Ethics:**

Ethics
Especially for datasets, are there any ethical concerns that warrant further discussion or review? See ethics guidelines on https://neurips.cc/public/EthicsGuidelines.

===

No, there are none. Data harvesting and many other aspects of data acquisition are free from ethical concerns. Data used for dataset creation is based on materials openly available for citizens and the public anyway. There is little to no potential for unethical use of dataset and consequences.

**Relation To Prior Work:**

Relation To Prior Work
Is it clearly discussed how this work differs from previous contributions?

===

Yes, it is discussed and that is onlyone dataset of this kind for Greek language.

**Summary And Contributions:**

Summary And Contributions
Briefly summarize the submission and its contributions. If desired, add formatting using Markdown and formulas using LaTeX. For more information see /faq

===

Authors demonstrate how newly introduced dataset of diachronic Greek language with Parliamentary proceedings was created via crawling, parsing, cleaning, processing and verifying data along the way. They discuss obstacles and deciisions made in the process.

With newly introduced dataset authors demonstrate how it could be used to perform various experiments and extract valuable insights in socio-political and lingustic domain, with a few interesting observations, proving dataset usability.

---

> ### Author Response · Authors · 2022-08-29
> **Response**
>
> Thank you for reviewing our work.
>
> Regarding your concern about the selection of Apache Tika for the conversion of the doc, docx and PDF files to text format, the main criteria according to which we selected this tool were 1) the coverage of file types and 2) the effectiveness of the tool to parse them, evaluated with multiple manual checks of the results. From the originally downloaded files, 2227 were already in text format, 2115 were in doc format, 882 were in docx format, and only 257 were in PDF format. From the 257 PDF files, 5 PDF files included images of scanned text and required optical character recognition, which we decided not to perform. Another 121 PDF files were problematic and the contents could not be decoded by any of the valid Greek encodings (note that there exist several different encodings for the Greek language). We excluded these 126 PDF files from our analysis as we considered their size small, compared to the other 5355 files that we had at our disposal.
>
> Concerning your interest in how different values of the frequency cut-off parameter could influence the results, we further evaluated the Compass model with different values of frequency cut-offs on the most frequent words for the task of detecting word usage changes between the decades 1990-1999 and 2010-2019. Specifically for each different random seed in the range [0-9], and for each frequency cut-off c, where c ∈ [0,50,100,200,250,300,400], we collected the top-k most changed words, where k ∈ [10, 20, 50, 100, 200, 500, 1000]. We plotted the stability of the different Compass variations and present the results in Figure 4 of the supplementary material. We present the discussion below:
>
> [lines 343-357]-> “In addition, we studied the effect that different frequency cut-offs have on the Compass variation. We implemented the Compass variation with different values of frequency cut-offs on the most frequent words for the task of detecting word usage changes between the decades 1990–1999 and 2010– 2019. Specifically for each different random seed in the range 0–9, and for each frequency cut-off c, where c ∈ [0, 50, 100, 200, 250, 300, 400], we collected the top-k most changed words, where k ∈ [10, 20, 50, 100, 200, 500, 1000]. For all Compass variations in this study, we removed words that appear less than 200 times. Fig 4 presents the results of the stability study. For k > 100, the results remain consistent and show that as the number of most frequent words rises, the stability of the approach decreases. For k < 100, the results do not show a consistent behavior. For c < 50 and c > 250, the intersection@k measure is lower, indicating lower stability. However, the implementations with c ∈ [100, 200, 250] present higher stability. These results imply that there is room for future studies on the effect of different frequency cut-offs on the stability of various semantic change detection approaches. For example, it would be interesting to investigate how the frequency cut-offs of the least frequent words might also affect the stability of an approach, either separately or when combined with different frequency cut-offs of the most frequent words in a corpus.”
>
> Finally, thank you for pointing out that the counter of downloads on Zenodo was 0. We contacted the Zenodo support department and they confirmed that the platform experiences delays in the statistics of the uploads. We are glad to note that this is not an issue anymore, as the counter has been updated.

---

### Review · Ethics_Reviewer_deRe · 2022-08-19

**Recommendation:** 1

**Ethics Documentation:**

The authors provide sufficient detail on the collection procedure to help with the transparency of the data collection method. The CC Attribution 4.0 International is fine for this data on zenodo.



**Ethics Review:**

The paper collects text records of parliamentary speeches in Greece. Since the original data is publicly available, and the individuals are speaking in a public forum, there are no issues about the sensitivity of de-anonymization of study participants. However, I am unsure about how to interpret GDPR requirements with respect to politicians requesting to be forgotten.

One suggestion for the authors is to discuss the difference in selection bias this data has, with respect to other corpora of Greek text. This is to foreshadow potential users in future who may ignore the fact that it is a very specific kind of text source and use it as a general purpose text representing semantic shifts.

---

> ### Author Response · Authors · 2022-08-29
> **Response**
>
> Dear Ethics Reviewer,
>
> Thank you for taking the time to evaluate our work. As per your recommendation, we have added in Section 6 “Conclusions and Future Work” the following statement:
>
> [lines 340-341] -> “Our dataset has a specific provenance, parliamentary recordings, and is not necessarily representative of language use and evolution in general.”
>
> The primary data for our dataset are in the public record, as the official parliamentary proceedings of the country. That said, we are of course committed to consider any reasonable and lawful request, if and when it comes, to amend or remove entries. We included this statement in Section 7 “Dataset License and Maintenance” of the supplementary material.
>
> Kind regards,
>
> The authors

---

### Author Response · Authors · 2022-08-29
**Thanks to Reviewers**

We would like to take the opportunity to thank the reviewers for their thoughtful comments, which we considered seriously and led to significant improvements in our work. As you can see in the detailed per-reviewer comments, we have made several clarifications, enriched related work, reran the experiments where needed and improved the presentation, responded to questions and added more details in the supplementary materials. We hope that these will address your comments and of course we welcome any further feedback you may have. Thanks again.

---

### Meta-Review · Area_Chair_Bhmm · 2022-09-09

**Recommendation:** Accept
**Confidence:** 4

**Metareview:**

The authors have compiled a new dataset using data from the Greek parliament. The dataset is interesting because it is comprehensive data (over 30 years) in modern Greek. It is also a political speech database that accounts for modern greek politics. This data will be helpful for people working on NLP to test their models and political scientists working on political discourse.

Some of the reviewers (and me) complain about some of the preprocessing that the authors have done, like attributing sex to speakers or removing stop words, which are derivative results that obscure the quality of the data. Still, the authors have responded to the authors' criticism, and the reviewers have acknowledged the responses by the authors positively.

---

### Decision · Program_Chairs · 2022-09-16

Accept